# Neural Contributions of the Hypothalamus to Parental Behaviour

**DOI:** 10.3390/ijms22136998

**Published:** 2021-06-29

**Authors:** Chitose Orikasa

**Affiliations:** Laboratory for Morphological and Biomolecular Imaging, Nippon Medical School, Sendagi 1, Bunkyo, Tokyo 113-8602, Japan; orikasa@nms.ac.jp

**Keywords:** MCH, parental behaviour, nursing, oxytocin, GABA

## Abstract

Parental behaviour is a comprehensive set of neural responses to social cues. The neural circuits that govern parental behaviour reside in several putative nuclei in the brain. Melanin concentrating hormone (MCH), a neuromodulator that integrates physiological functions, has been confirmed to be involved in parental behaviour, particularly in crouching behaviour during nursing. Abolishing MCH neurons in innate MCH knockout males promotes infanticide in virgin male mice. To understand the mechanism and function of neural networks underlying parental care and aggression against pups, it is essential to understand the basic organisation and function of the involved nuclei. This review presents newly discovered aspects of neural circuits within the hypothalamus that regulate parental behaviours.

## 1. Introduction

Maternal behaviour is a distinct sex-related factor in mammalian reproduction. Females exhibit maternal care after parturition, while males who encounter pups engage in infanticide [1,2,3]. These behaviours depend on sexually dimorphic features of the brain shaped by the effects of gonadal steroid hormones [4,5,6,7] and sex chromosomes [8,9,10]. It is also proposed that epigenetic modifications, i.e., DNA methylation and histone acetylation may regulate gene expression associated with brain sexual differentiation [11]. Brain differentiation between the sexes occurs early in development, during the so-called ‘critical’ period, leading to differences in neural circuits, endocrine systems and behaviours [12,13] that persist throughout the life of the animal. Gonadal steroids act on the molecular and cellular levels to influence the neural structure and function of the brain. Males are exposed to testicular steroids during this critical neonatal period, resulting in brain masculinisation. In the absence of testicular steroids, the brain is feminised. The differences in sex-dependent reproductive behaviour are assumed to result from these differences in exposure to gonadal steroids during the critical period. Parental care is a reproductive behaviour that can change even in adults in response to alterations in the endocrine milieu or social impetus. In rodents, parturient females display maternal care; virgin females, who are less interested in pups and maternal care, are easily motivated after priming with several exposures to pups [1]. While virgin males sometimes engage in infanticide [3], males that are mating with gestating females exhibit parental behaviour [2,3]. Hormonal circumstances change in adult females, dynamically altering serum oestrogen levels. In female mice, inhibition of oestrogen receptor α in the medial preoptic area results in the absence of maternal behaviours [14]. However, oestrogen replacement therapy in adult male mice has no effect on their parental behaviours (unpublished data). Therefore, the administration of hormones is not sufficient to induce parental behaviour.

In the author’s previous study, the social isolation of virgin mice induced parental behaviour in both sexes [15]. In addition, changing the social context has consequences on certain parental behaviours, such as males exhibiting parental nursing behaviour or females ignoring pups. Although sex-dependent behaviours arise from differences in brain differentiation, these behaviours are presumably open to alteration by social stimuli. These results suggest that the neuronal pathways involved in parental behaviour retain a high proportion of plasticity, even in adults.

## 2. Parental Behaviour in Male Mice

Male mice who have mated and then cohabitated with gestating and delivering females have been observed to repress attacking pups and to exhibit parental behaviour [2]. We previously observed that parental behaviour in both virgin male and female mice was induced by a very long period of social isolation. Social isolation prompts parental behaviour in both sexes [15]. Studies have reported that social isolation can be a stressful situation in rodents [16,17,18,19,20] and arise as a result of various behavioural changes, i.e., enhanced aggression [19], depression-like behaviour [20] and levels of impulsivity [19]. Moreover, social isolation changes various behaviours, such as aggressive or depression-like behaviour [21,22]. Aggression using the resident-intruder test [22] revealed that single-housed male mice showed more aggressiveness towards the intruder male mice than the group-housed mice [23]. It is still controversial that isolation stress enhances aggressive behaviour; however, it reduces the attacking of pups. These results suggest that neural circuits in these events differ because of the distinct functional significance in social behaviour. The synaptic machinery of the brain circuits involved in parental behaviour change in response to social conditions. Social isolation in animals and humans is considered as an intensive stressor, which impairs learning. Social isolation is thought to induce changes in social behaviour by inducing neuroanatomical changes that alter the function of the neuroendocrine system. Neuronal plasticity and synaptic remodelling of the nervous system are retained in adulthood under certain conditions, such as isolation stress [16,17,18,19,20,21,24,25,26]. In our previous study on *MCH-tTA*; *TetO DTA* bigenic mice, +/+ bigenic virgin males with abolished MCH neurons were more aggressive towards pups presented as well as intruder males than the +/− controls. Therefore, the possible involvement of neural circuits for aggressiveness towards pups and intruder males is identical to that of responsiveness, including the MCH neuronal activity. Social isolation elicits presynaptic remodelling in the nucleus accumbens (NAc) neurons, including synaptic plasticity in emotional behavioural responses [27], and changes the synaptic neurotransmission of receptor subtypes in the dorsal raphe nucleus, resulting in altered neuroplastic connectivity regarding social rewards [28].

## 3. Evidences of Neuromolecular Regulation of Parental Behaviour

The medial preoptic area (mPOA) [1,2,3] and anteroventral periventricular nucleus [29] are critical components of the neural system governing parental behaviour. Candidate regulators of parental behaviour include neuropeptides galanin [30] and oxytocin (OT) [31,32]. Tyrosine hydroxylase is involved in maternal behaviour in females but not in males [29]. Galanin is expressed in the mPOA neurons, which are activated in both sexes by parenting episodes involving pup grooming and retrieval behaviour [3]. OT-secreting neurons in the paraventricular nucleus (PVN) play a crucial role in the onset and maintenance of maternal behaviour in rodents. OT is subjected to nursing and facilitated parental behaviour [32,33], and it then becomes feasible in participating with the auditory cortex in responding to pup calls [31]. In humans, OT release [34] and OT itself improved parenting [35] in terms of the formation of social memory [36,37].

## 4. Involvement of MCH Neurons in Parental Care

Projections from the PVN posterior to the lateral hypothalamic area (LHA) regulate the melanin concentrating hormone (MCH) neurons [38], a neuromodulator that integrates physiological functions [39,40,41,42,43,44,45,46,47,48,49,50,51,52,53,54,55]. Neural projection from OT neuron in PVN to MCH neuron in LHA [38], which expresses OT receptor [38,50], is involved in mating, parenting and social cognition. Moreover, the MCH receptor (MCHR) [51] is distributed throughout the area that regulates reward including the NAc [47]. The MCHR distribution correlates with oxytocinergic projection and may be involved in the emotional reinforcement of rewards [48].

The MCH, a 19-aminoacid cyclic peptide, was first characterised in salmon pituitary extracts as a circulating factor that mediated colour changes in teleost fishes [56]. MCH is distributed in the lateral hypothalamus, dorsomedial hypothalamus and zona incerta [57]. In mammals, MCH neurons play a crucial role as neuromodulators that integrate physiological functions involving energy balance [39,40,41], sleep [42,49,50], olfaction [43], anxiety [44] reward [45,46,47,48], and cognition [48,49]. The ablation of MCHR affects maternal behaviours, especially impaired retrieving pups and increased attacking pups [58].

## 5. Effect of Congenital Ablation of MCH on Nursing Behaviour

Higher expression of the immediate early gene *c-fos* in the MCH neurons was observed in virgin female and male ddN mice that showed nursing crouching behaviour than in those that ignored their pups after social isolation [59]. To determine the neural rudiment governing nursing behaviour, studies have been conducted in MCH-neuron knockout animals, such as *MCH-tTA*; *TetO diphtheria toxin A fragment (**DTA)* bigenic mice [59] using the tet-off system (Figure 1). MCH neurons are specifically ablated in *MCH-tTA*; *TetO DTA* +/+ bigenic mice, with the orexin neurons intact. The bigenic *MCH-tTA*; *TetO DTA* +/+ bigenic female mice had a lower pup survival rate than did *MCH-tTA*; *TetO DTA* +/− bigenic controls. The body weight of *MCH-tTA*; *TetO DTA* +/+ bigenic mice was significantly lower in both sexes because of the physiological role of MCH neurons in food intake [39] and energy metabolism [40,41]. No difference in food intake (Kcal/day) was observed between MCH knockout mice and wild type controls [60]. The locomotor activity of MCH knockout mice is significantly elevated as compared with controls, resulting in reduced weight gain as a consequence of increased energy expenditure [59]. The virgin *MCH-tTA*; *TetO DTA* +/+ bigenic females display less maternal care in regard to crouching behaviour comparable to that of *MCH-tTA*; *TetO DTA* +/+ bigenic mothers (X^2^ = 11.29, df = 1, *p* = 0.001), whereas virgin +/+ bigenic males exhibit aggressiveness toward their pups (Figure 2). However, no significant difference in retrieving behaviour was observed. Together, these findings indicated that the MCH neurons play a pivotal role in parental nursing behaviour in mice.

## 6. Effect of Optogenetic Stimulation Intensity on Behaviour

The nursing crouching behaviour was elicited by low-frequency (473 nm, 10 ms, 0.5 Hz, 1 mW) photo-stimulation through the optic fibres present in both sexes of channelrodopsin 2 (ChR2)-expressing *MCH-Cre* mice [59]. In contrast, no parental behaviour was observed, in response to applied laser pulsed similar to the condition of rapid eye movement (REM) sleep (475 ± 17.5 nm, 2.5 mW,10 ms, 10 Hz) [42]. The condition of the parental nursing behaviour was a low-frequency stimulation (473 nm, 10 ms, 0.5 Hz, 1 mW), whereas that of REM sleep induction was a high frequency stimulation (475 ± 17.5 nm, 2.5 mW,10 ms, 10 Hz). ChR2-positive MCH neuron was denoted *c-fos* expression in the condition of animals exhibiting parental nursing behaviour. In animals exhibiting crouching behaviour, about 40% of the MCH neurons expressed ChR2, and 10% of the ChR2-positive MCH neurons expressed *c-fos* [59]. A previous study of the ventromedial hypothalamus (VMH) showed that mounting and attacking intruder males were elicited by different intensities of the oestrogen receptor 1 (Esr1) in the ventromedial nucleus [61,62]. Optogenetic induction of attack requires the presence of more Ers1 cells containing ChR2 than does the induction of mounting behaviour. Therefore, optogenetic stimulation might coordinate the threshold activity more robustly and coincide with tuned cells, inducing either attack or opposing other behaviours. The number of Esr1 cells expressing ChR2 and *c-fos* was much higher in case of induction of attack than that in case of mounting behaviour. These findings suggest that each behaviour is dependent on the frequency and intensity of photo-stimulation. Therefore, the type of behaviour elicited depends on the responsiveness of the neurons regulating the particular behaviour. Sensory cues, required for activating distinct neuronal populations in the same nucleus with different thresholds, are responsible for determining specific behaviours. The strength of the optogenetic photo-stimulation corresponds to signals from the accumulation of olfactory, auditory, haptic, visual and environmental cues.

Some of the neurons in the VMH, involved in mating, attacking or both are responsive to appropriate stimulation in the nucleus. Low-intensity signals induce mating, while high-intensity signals elicit attack behaviour [62]. We ask the question: Why do the functional differences in the same nucleus lead to behaviour differences, and how do neurons in the hypothalamus convey sensory information to induce inherent behaviours? The MCH neuronal state for behaviour is variably induced by photo-stimulation: Low stimulation elicits nursing, and high stimulation elicits REM sleep. The neural circuit responsible for this MCH-induced behaviour could communicate with other brain areas to integrate each behaviour. The extraordinary event of the behaviour must be induced by different reactivities of the responsible neurons.

## 7. MCH Neural Relay in PVN Oxytocin Neurons Is Involved in Nursing Behaviour

Further evidence shows that MCH neurons are regulated by OT neurons in PVN that project anatomically posterior to LHA [38]. OT, a neurotransmitter synthesised in both the PVN and supraoptic nuclei of the hypothalamus [63], regulates peripheral reproductive-related functions and central actions in the brain. OT secretion from the posterior pituitary gland induces uterine contractions during parturition and also acts on the muscle in the mammary gland trabeculae to induce milk ejection during lactation [63]. Oxytocinergic neurons are involved in a variety of central nervous system functions. Centrally and peripherally secreted OT acts through the OT receptor. This receptor is distributed in the ventral tegmental area (VTA) and NAc and is involved in feeding, sexual behaviour and reward properties of social interactions and the formation of social bonds. OT has been shown to facilitate the onset of maternal behaviour in rodents [31,32]. The possible mechanism responsible for parental nursing behaviour is involved in the neural relay for the LHA-PVN within the hypothalamus. Studies have shown that the periaqueductal grey (PAG) in the midbrain is implicated in reproductive behaviour such as the females’ lordosis behaviour and the maternal arched back crouching behaviour, whereas no effects were recorded in the pup grooming behaviour [64,65,66].

Alternatively, the stimulation of the projection to the PAG from the galanin neurons results in pup grooming, albite with no effect on crouching in both sexes [30]. Maternal behaviour could be regulated by the LHA–MCH neuronal input to the PAG. GABAergic neurons in the LHA-to-PAG projection precipitate in predatory hunting in mice [66]. Whereas, the role of the MCH receptor in the PAG is yet to be determined. However, the method whereby the neural circuit for these diverse parenting behaviours govern each of the behavioural contents is still controversial.

## 8. OT Enhances the Neural Circuits of Rewarding from Pups

Previous studies have identified the mPOA as a critical region in the regulation of parental behaviour [1,2,3,30,67,68,69,70]. OT neurons act on mPOA, VTA and NAc to prompt parental behaviour. OT neurons in PVN receive the projection from the LHA [59]. In our previous study, *MCH-tTA; TetO DTA* bigenic (+/+) female mice with the complete innate ablation of MCH neurons displayed less attention towards pups and less maternal care than *MCH-tTA*; *TetO DTA* +/− bigenic controls, which was similar to *MCH-tTA*; *TetO DTA* +/+ bigenic mothers, that display significantly lowered crouching than the +/− controls. Moreover, the virgin +/+ bigenic females showed significantly lowered crouching than *MCH-tTA*; *TetO DTA* +/− bigenic controls. MCH neurons are ablated partially using Cre recombinase-dependent DTA, which abolishes approximately 73% of MCH neurons in virgin females. Virgin females with partially ablated MCH neurons exhibit crouching behaviour for less time than green florescent protein controls. The MCHR expression is necessary to the reward circuitry of the NAc as is the anatomic integrity of the oxytocinergic projection of the mesolimbic system; these findings indicate a possible alliance between these factors in the emotional reinforcement of rewards for parenting. The MCHR is expressed in the olfactory regions, neocortex, hippocampus, NAc, amygdala, ventromedial hypothalamic nucleus and locus coeruleus. OT receptor expression [71], coupled with MCHR in the NAc has a synergistic effect on inherent rewarding, contributing to the execution of parental behaviour [72]. Therefore, MCH neural networks along with OT signalling in reward circuitry facilitate pup survival.

Maternal rewards system may contribute to maternal nursing systems. Relationships between maternal depression and OT levels were demonstrated previously [73]. MCH–LHA projects closely to OT–PVN, which is the stimulation that induces parental behaviour along with increasing plasma OT levels [59]. The recurrent process between the PVN projection to LHA and the LHA projection to PVN are assumed to be tuning properties for continuous parenting crouching behaviour. Almost all MCH neurons expressed OT receptor mRNA; however, OT neurons faintly expressed the MCH receptor [38,50].

## 9. Social Isolation Modifies GABAergic Transmission

Extended periods of social isolation can affect parental behaviour by inducing neuroanatomical changes. The expression of the immediate early gene *c-fos* in the MCH neurons increased during parental nursing behaviour of mice after social isolation [59]. MCH neurons in the LHA contain and release γ-aminobutyric acid (GABA) [59,74] as well as express GABA-synthesising enzymes GAD65 and GAD67 [59,75,76]. MCH neurons also contain and release glutamate in the lateral septum [77,78]. In the amygdala neural circuits, GABAergic and glutamatergic neurons in the VTA specifically tune each rewarding and aversive motivational predicaments [79]. mPOA, which is indicative of GABAergic neurotransmission, governs parental behaviour, whereas glutamatergic neurons in the same nucleus are associated with anxiety-like behaviour [80]. Glutamatergic neurons in the mPOA regulate anxiety-like behaviour, while GABAergic neurons contribute to anxiolytic effects, i.e., parental behaviour, indicating that the mPOA in the same nucleus plays a crucial role in reconciliation of opposite behaviours. Moreover, the mPOA projections to the neurons in midbrain reward circuits may prompt parental behavior with accommodation for dopamine release [80]. Therefore, GABAergic and glutamatergic neurons play a prominent role in opposing effects on the social behaviour. These results suggest that the same nucleus governs opposite positive or negative behaviours by discriminating the neurotransmission of the nucleus. Social isolation, which prompts parental behaviour for several resting weeks, could change the statement of the brain neurotransmission. In fact, conditions, such as accrues to excitatory neurotransmission of GABA neurons result in a profound depolarising shift in magnocellular neurosecretory cells that secrete OT in the PVN [81]. Our previous study has shown that MCH fibre expressing enhanced yellow fluorescent protein projected close proximity to the OT neuron in the PVN. The MCH neuron was revealed to express GABA, which innervates as a neurotransmitter and forms synapses with OT neurons [59]. Moreover, GABA agonist musicimol injected into the PVN increases *c-fos* in the OT neurons. More *c-fos* expressing OT neurons were observed in the socially isolated female and male mice than the co-habituated female and male mice [59], indicating that the MCH neuron could regulate excitatory OT neurons in the PVN. Although GABA principally functions as an inhibitory neurotransmitter in the brain, excitatory GABAergic activity is identified in MCH neurons during development [82]. In mature neurons, but the excitatory action of GABA under stress conditions has been elucidated [83,84,85,86]. Social isolation might be presumed to be a validated stressor and to elicit changes in the synaptic organisation action in the rodent brain [16,28].

Therefore, social isolation stress may change the mode of GABAergic excitation. The projection from LHA to PVN under social, reward-context associations are responsible for the LHA–PVN-evoked OT releases implicated in parenting augmentation. Therefore, social isolation may change the mechanisms underlying the modality of GABAergic excitation. Abolishing MCH neurons may induce the superiority of glutamatergic circuit, thereby, stimulating broad area commitment to infanticide in the brain. Moreover, the impairment of OT neurons showed acceleration aggression. The balance between GABA and glutamate utilisation in the MCH neurons in some aspects of parental and opposing behaviours remains to be elucidated.

## 10. Aggressive Behaviour towards Pups

Pheromonal signals are received by neurons in the vomeronasal organ (VNO) [87] and the main olfactory epithelium within the nasal cavity [88]. In rodents, olfaction is known to be important for the identification of conspecifics and sex differentiation. The excision of the olfactory bulb results in defects in aggressive behaviour, indicating that olfactory perception is involved in dictating aggressiveness. A body of evidence implicates neural networks in the governing of aggressive behaviour, including a social behaviour circuit involving the mPOA, medial amygdala (MeA), bed nucleus of the stria terminalis (BNST) [89], lateral septum, anterior hypothalamus, VMH and PAG [65]. Social signals detected by the olfactory bulb are subsequently transmitted to specific brain regions: MeA and then to the VMH or BNST [90]. The VMH is downstream of the MeA, which in turn disinhibits the VMHvl glutamatergic neuron induction of aggressive behaviour; for example, this circuitry guides the behaviour in which male mice attack male intruders but not females [91,92]. Distinguishing between females and males is also accomplished through the detection of semiochemicals, some of which are major urinary proteins [93,94]. These chemicals are detected by sensory neurons in the VNO in a sex-dependent manner [95,96,97]. For example, during infanticide by virgin male mice, pheromonal signals from pups via the VNO are sent to the accessory olfactory bulb, then to the MeA and relayed to the anterior hypothalamic area/VMH and BNST. The VNO neurons presumably are involved in the detection of pheromonal signals related to parental care [3]. Knockout of cation channel subfamily C member 2 (*Trpc2*) in male mice causes impaired VNO-input signalling, resulting in reduced attacking of pups, indicating that VNO signalling elicits the attacking of pups [98,99,100]. Our previous study reports that *Trpc2* KO mice spent more time licking the pups and crouching after social isolation. However, retrieval behaviour increased only in response to social isolation and was not affected by *Trpc2* KO. These results indicate that not all the social signals are transmitted by the VNO [101].

The *MCH-tTA*; *TetO DTA* +/+ bigenic virgin males with ablated MCH neurons were more aggressive toward the pups. In the resident-intruder test, the *MCH-tTA*; *TetO DTA* +/+ bigenic virgin mice, exhibited more male–male aggression than did *MCH-tTA*; *TetO DTA* +/− bigenic controls [59]. Ablation of MCH neurons also leads *MCH-tTA*; *TetO DTA* +/+ bigenic male mice to exhibit more aggressiveness against other male mice and pups, suggesting that MCH neurons disinhibit the olfactory circuit and sensory integration from the olfactory bulb. This result is similar to that observed with neural circuit modulation that results in male attack on pups and intermale aggressive behaviour [102]. The MeA is evidenced as an inhibitor of maternal behaviour [103,104,105,106,107]. Moreover, MCHR in the MeA is assumed to be implicated in maternal aggression towards intruder male [107]. The VNO and main olfactory bulb may be substantially involved in male–male aggressive behaviour prompted by pheromones [100] and play decisive roles in conferring infanticide in mice. The *MCH-tTA*; *TetO DTA* +/+ bigenic virgin mice were able to mate because of their ability to discriminate sex due to their preserved VNO function. Neural circuits for attacking pups may be involved in relaying from MCH neurons in the LHA (MCH–LHA) to OT neurons in the PVN (OT–PVN) in rodents [59].

The ablation of MCH neuron excitability to PVN may prevent the induction of OT secretion. In contrast, *MCH-tTA*; *TetO DTA* +/+ bigenic female mice ignore the pups, indicating that the effect of abolishing MCH neurons differed in some degree between females and males. OT transmits signals involved in social interactions, such as parental and pair bonding; abolishing OT facilitates aggressive behaviour [108]. Presumably, OT acts on the mPOA, which induces parental behaviour in rats [109,110].

Another proposed mechanism is effects on OT, which acts to regulate the salience of external social cue rather than affiliative behaviours [111,112,113], which is a critical role of OT in the event of the discriminate mode of an emotional action in a conspecific [114].

## 11. Parental Licking Behaviour

Optogenetically-evoked crouching behaviour requires around 10% of the ChR2-expressing and MCH cells that express *c**-fos*. The photo-stimulation of ChR2 MCH neurons significantly increased crouching behaviour, but did not affect licking behaviour [59]. Therefore, the relative contribution of MCH neurons to licking behaviour may be minimal. Mice contact pups at first and pup-licking is assumed to be dictated by emotive excitation or apprehension [115,116].

Optogenetic stimulation of galanin neurons in the mPOA induces retrieving and pup grooming rather decreasing attacking pups [30] and had no effect on other parental behaviours. Genetical ablation of galanin neurons in the mPOA induces pup attacking in virgin females but not in mating-experienced females and males. Therefore, the brain centre for pup retrieving and grooming behaviour is in the mPOA, and galanin is one of the molecules involved in parenting pup grooming. Pup grooming behaviour is also affected by GABAergic neurons in the posterodorsal (MeApd) in females [78]. The effect of photo-stimulation on retrieving pups and crouching is less than that on pup grooming. Higher GABAergic neuron activity in the MeApd induces the attacking of pups, while low activity of these neurons prompts parenting in male mice. Opposing behaviours, such as parenting and aggression, are centred in different regions of the brain. For example, MeApd facilitates parenting behaviour, while aggression is colinear with the quantitative responses of GABAergic neurons in the brain.

## 12. Conclusions

Parental behaviour is composed of sequential behaviour events induced by an associated nucleus for each event that is stimulated by a social cue. In this chapter, I proposed that the key brain regions and molecules involved in regulating parental behaviour reside in the POA, the focus of much research on this topic. Genetic ablation of MCH neurons in transgenic, *MCH-tTA*; *TetO DTA* +/+ mice results in impaired nursing parental behaviour. Virgin *MCH-tTA*; *TetO DTA* +/+ bigenic males engaged in infanticide toward the pups, while females ignored pups. A neural circuit from the LHA–MCH to PVN–OT was revealed in this study, and MCH reward neural circuitry, together with OT signalling, is a requisite for parental behaviours that promote pup survival.

## Figures and Tables

**Figure 1 ijms-22-06998-f001:**
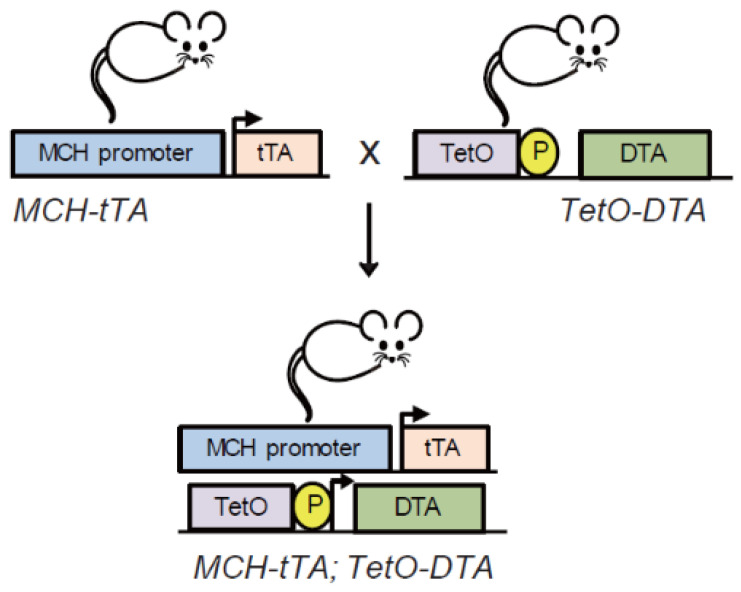
Generation of *MCH-tTA; TetO DTA* bigenic mice. Cell-specific stratagem of innate MCH neurons ablation observed congenitally in the bigenic mice. Tetracycline-controlled gene expression and tTA-induced DTA ablation observed specifically in MCH neurons (Modified from Tsunematsu et al. [42]).

**Figure 2 ijms-22-06998-f002:**
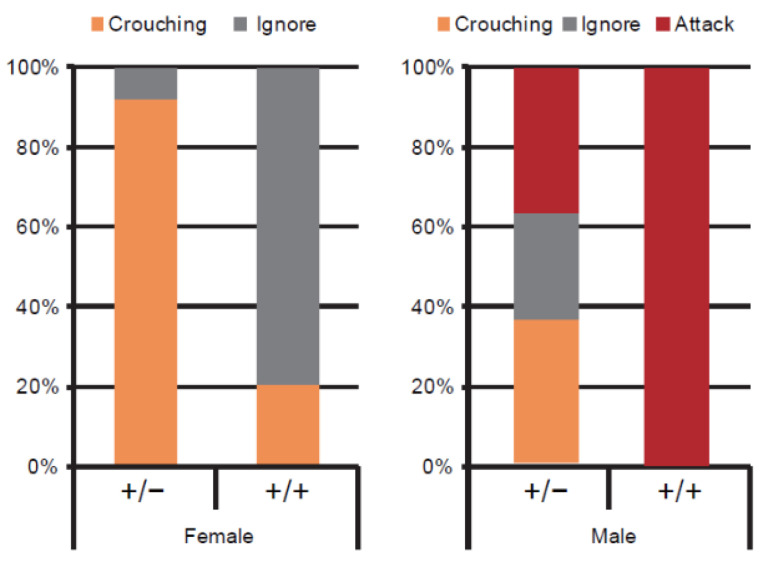
Cell-specific ablation of MCH neurons impairs nursing behaviour in both virgin female and male mice. Percentage of parental behaviour or attack pups of virgin *MCH-tTA*; *TetO DTA* bigenic. +/+ and +/− female (**left**) and male (**right**) mice. The ratio of parental behaviour (crouching) in female mice was analysed by Chi-squared test; X-squared = 9.56, df = 1, *p* = 0.002 (From Kato et al. [59]).

## Data Availability

Not applicable.

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
