# Peer review of "Neural Contributions of the Hypothalamus to Parental Behaviour"

_ijms, 2021, doi:10.3390/ijms22136998_

Round 1

Reviewer 1 Report

The work is well written and structured. The sections are clear and describe in detail the involvement of MCH in the alterations of the behavior of parental care. Also, the description of different research approaches and of the results obtained  from behavioral investigation using many protocols  resume clearly the body of evidences about the specific field of interest.

I suggest to accept the work in the present form.

Author Response

I appreciate the valuable comment made by reviewer 1.

I trust that the revised manuscript is now ready for publication.

Reviewer 2 Report

This review is a remarkable attempt trying to understand the behavior of nuclei, neural circuits and the involved neurotransmitters, particularly the neuropeptide MCH, in the complexity of parental behavior. The author proposes that “…the key brain regions and molecules involved in regulating parental behaviour reside in the POA…A neural circuit from the LHA–MCH to PVN–OT is revealed in this study, and MCH reward neural circuitry together with OT signalling is a requisite for parental behaviours that promote pup survival”.

I have several observations and doubts:

Please, check grammar, typos and sentence construction throughout the text. See for example lines: 31-32, 57, 93-94, 175-176, 200-201, 235-236…

When mentioning brain nuclei, neural circuits, neurons, and neurotransmitters, the construction of sentences is unclear and can lead to confusion. For example, the construction of sentences when using MCH, MCH neurons, OT, OT neurons, the different nuclei which are cited as PVN, LHA ..., neural circuits… is not clear. For example, in the sentence on lines 93-94: When you say: “Neural circuit from OT in the PVN to MCH in the LHA” you mean: The neural connection between PVN and LHA involve OT and MCH?

A schematic diagram between the nuclei and neurotransmitters that you mention, could help to understand the complex mechanisms that underlie the parental behavior.

Do MCH neurons only contain MCH or does it coexist with other neuromodulators?

GABA is an inhibitory neurotransmitter and glutamic an excitatory one. However, in the text (point 9), their inhibitory or excitatory role is not clear.

In paragraph 164-173, what nuclei and areas are involved?

Is the connection between PVN and LHA bidirectional? Please, specify.

TyrOH is mentioned in the text (line 82). Are catecholamines involved in parental behavior?

Author Response

I appreciate the valuable comments made by reviewer 2. I have revised the manuscript accordingly, as itemized below. I trust that the revised manuscript is now ready for publication.

This review is a remarkable attempt trying to understand the behavior of nuclei, neural circuits and the involved neurotransmitters, particularly the neuropeptide MCH, in the complexity of parental behavior. The author proposes that “…the key brain regions and molecules involved in regulating parental behaviour reside in the POA…A neural circuit from the LHA–MCH to PVN–OT is revealed in this study, and MCH reward neural circuitry together with OT signalling is a requisite for parental behaviours that promote pup survival”.

I have several observations and doubts:

Please, check grammar, typos and sentence construction throughout the text. See for example lines: 31-32, 57, 93-94, 175-176, 200-201, 235-236…

Answer: I thank the reviewer for pointing out this. All the text of the manuscript has been checked by proofreading service. According to the advices of proofreading service, I have rewritten the sentences: P3, L16 - L17; P5, L5 - L6; P7, L6 - L8; P11, L7 - L8; P12, L15 - 16; P14, L13 - L16 corresponding for example lines: 31-32, 57, 93-94, 175-176, 200-201, 235-236, respectively.

When mentioning brain nuclei, neural circuits, neurons, and neurotransmitters, the construction of sentences is unclear and can lead to confusion. For example, the construction of sentences when using MCH, MCH neurons, OT, OT neurons, the different nuclei which are cited as PVN, LHA ..., neural circuits… is not clear. For example, in the sentence on lines 93-94: When you say: “Neural circuit from OT in the PVN to MCH in the LHA” you mean: The neural connection between PVN and LHA involve OT and MCH?

Answer: I’m sorry for the confusing description. I have rewritten the sentence: P7, L6 - L7.

A schematic diagram between the nuclei and neurotransmitters that you mention, could help to understand the complex mechanisms that underlie the parental behavior.

Do MCH neurons only contain MCH or does it coexist with other neuromodulators?

Answer:  I thank the reviewer for this comment. I have rewritten the sentences: P14, L9 - L11.

GABA is an inhibitory neurotransmitter and glutamic an excitatory one. However, in the text (point 9), their inhibitory or excitatory role is not clear.

Answer:  I apologize for this vague description. I have rewritten the sentences: P15, L16 - P16, L2.

In paragraph 164-173, what nuclei and areas are involved?

Answer:  I have rewritten the sentences: P10, L13.

Is the connection between PVN and LHA bidirectional? Please, specify.

Answer:  I thank the reviewer for this comment. I have rewritten the sentences: P12, L15 - L16 & P13, L19 - P14, L3.

TyrOH is mentioned in the text (line 82). Are catecholamines involved in parental behavior?

Answer:  I thank the reviewer for pointing out this. I have rewritten the sentences: P14, L19 - P15, L2.